# A Challenge for Palliative Psychology: Freedom of Choice at the End of Life among the Attitudes of Physicians and Nurses

**DOI:** 10.3390/bs10100160

**Published:** 2020-10-21

**Authors:** Ines Testoni, Camilla Bortolotti, Sara Pompele, Lucia Ronconi, Gloria Baracco, Hod Orkibi

**Affiliations:** 1Department of Philosophy, Sociology, Pedagogy and Applied Psychology (FISPPA), University of Padova, 35131 Padova, Italy; cami.borto@gmail.com (C.B.); sara.pompele93@gmail.com (S.P.); 2Emili Sagol Creative Arts Therapies Research Center, University of Haifa, Haifa 3498838, Israel; horkibi@univ.haifa.ac.il; 3Statistical Services, Psychology Multifunctional Center, University of Padova, 35131 Padova, Italy; l.ronconi@unipd.it; 4Home and Palliative Care Department, ULSS n. 2 Marca Trevigiana, Asolo, 31011 Treviso, Italy; gloriaangela.baracco@aulss2.veneto.it

**Keywords:** Advance Healthcare Directives, euthanasia, healthcare professionals, palliative care, palliative psychology, death education

## Abstract

This article considers a particular aspect of palliative psychology that is inherent to the needs in the area of attitudes concerning Advance Healthcare Directives (AHDs) among Italian physicians and nurses after the promulgation of Law No. 219/2017 on AHDs and informed consent in 2018. The study utilized a mixed-method approach. The group of participants was composed of 102 healthcare professionals (63 females and 39 males). The quantitative part utilized the following scales: Attitudes toward Euthanasia, the Religious Orientation Scale, the Balanced Inventory of Desirable Responding, and the Testoni Death Representation Scale. The results were mostly in line with the current literature, especially concerning a positive correlation between religiosity and the participants’ rejection of the idea of euthanasia. However, the qualitative results showed both positive and negative attitudes towards AHDs, with four main thematic areas: “Positive aspects of the new law and of AHDs”, “Negative aspects of the new law and of AHDs”, “Changes that occurred in the professional context and critical incidents”, and “Attitudes towards euthanasia requests.” It emerged that there is not any polarization between Catholics or religious people and secularists: Their positions are substantially similar with respect to all aspects, including with regard to euthanasia. The general result is that the law is not sufficiently understood, and so a quarter of the participants associate AHDs with euthanasia. Discussions on the opportunity for palliative psychologists to help health professionals to better manage these issues through death education courses are presented.

## 1. Introduction

Palliative psychology is gaining increasing importance not only in the intervention in the care of terminal patients, but especially in the context of promoting positive relationships between the healthcare community and patients’ families [1,2]. One of the most complex aspects to take into account at the end of life is respect for the patient’s wishes, which, in some cases, correspond to the need to not undergo exaggerated and futile interventions or prolong a life that is perceived as not worth living [3]. With respect to the cultural context, Italian bioethical history has long been marked by a dysfunctional split between secular positions and the will of Catholic authorities [4,5,6,7]. The development and diffusion of palliative care and the recognition of the importance of paying psychological attention to the patient have significantly changed the cultural climate, so much so that on 31 January 2018, Law No. 219 (“Norms concerning informed consent and Advance Healthcare Directives”) became effective, marking a fundamental turning point in Italian palliative care practices. The law, containing regulations on informed consent for medical care, Advance Healthcare Directives (AHDs), and advance care planning, was approved by the Italian Parliament in December 2017, and it came into force on 31 January 2018. This law “protects the right to life, health, dignity, and self-determination of the person and establishes that no health treatment can be initiated or continued without the free and informed consent of the person concerned, except in cases specifically provided for by law”, in compliance with the principles of the Constitution (Arts. 2, 13, and 32) and the Charter of Fundamental Rights of the European Union. It also tackles other delicate aspects regarding these topics, such as pain therapy and deep sedation in palliative care, as well as minors and adults who are uncapable of giving consent. Thanks to this law, people and patients may prepare and sign specific documents through which they express their will concerning medical treatments, especially in cases of severe sickness and before or during the end of life (EOL). Indeed, prior to this change, healthcare professionals had no clear guidelines, and a general confusion characterized the EOL of many patients, who often submitted to aggressive and disproportionate treatments [8]. Excluding the possibility to obtain euthanasia or assisted suicide, after enactment of the law on AHDs in Italy, adults can present a written text in which they describe the kinds of treatments they want or do not want to receive, even including those that could save their lives. Specifically, it is now possible to both withhold and withdraw artificial nutrition and hydration, as well as ventilatory support.

In line with the palliative care model, the most important aspect of this law is the trust between patients and healthcare professionals in order to safeguard individuals’ self-determination [9]. Palliative psychology in this field may be very useful because it can help health professionals to better manage all the difficulties related to the decision-making processes while respecting the law and, at the same time, the patients’ will [2]. Indeed, nowadays, social law is at risk of being insufficiently respected, since the possibility of adequately applying AHDs also requires a certain amount of psychological preparation for both common people and healthcare professionals [10]. Among the factors that may reduce the complete application of this right are the cultural removal of dying processes due to the fear of death [11,12], as the Terror Management Theory widely explains [13], the following lack of competence that creates difficulties in EOL decision-making processes [14], and ideological concerns, such as the presence of rigid religious positions [15,16,17]. Both of these factors are related to the widespread Western tendency to remove any discourse on human mortality [18], as the literature has widely highlighted [19,20,21].

In this regard, in the last decades, growing attention has been given to this issue, particularly in the field of thanatology. Indeed, training concerning death, dying, and bereavement has been shaped, and the positive results have been well documented [22,23,24]. For example, in Italy, a recent study on the theme of AHDs involved both social work and psychology students and highlighted how participants benefitted from such a course and how their attitudes towards AHDs became more open and flexible [25].

Because of this, the theme of death education has been gaining more and more attention in the field of healthcare professions as well because it allows the creation of training courses that help people come to know and manage topics, experiences, and emotions related to death [26,27].

In particular, this type of intervention could be managed by palliative psychologists, who can prepare their colleagues in palliative teams in the field of doctor–nurse–patient relationships. This kind of intervention may improve relational skills in order to respect the patient-centered approach [28,29,30,31] promoted by the World Health Organization (WHO), which is necessary for authentic and effective communication [32,33,34].

Literature specifically focused on the way healthcare professionals manage AHDs shows the importance of competence in this field [35]. Masuda and colleagues [36], for example, highlighted that one crucial concern for physicians was the lack of knowledge that people had on the matter, as well as the consequent need for more precise education on this, together with the fear that family members of a patient unable to express his/her will might make a decision that is, in fact, contrary to the one the person would have made because of a lack of proper communication between the patient, his/her family, and the physician. Moreover, Coleman [37] added that the fear of legal problems, a strict approach to religion, a paternalistic approach to medicine, and the fear of a high number of euthanasia and assisted suicide requests because of the introduction of AHDs were also major obstacles for physicians. To the contrary, some elements that could encourage the application of AHDs are previous positive experiences and the participants’ respect for patients’ autonomy and desires [37].

In this field, some crucial insights have also been offered by studies in the nursing area, which have highlighted how the lack of knowledge and of proper training concerning the theme of AHDs is a serious and widespread criticality that might undermine nurses’ confidence in approaching the theme with patients [38,39]. A recent review on the implementation of AHDs has confirmed the importance of proper training on this theme, whose lack is considered particularly detrimental in all health professions [40]. Similar conclusions have also been reached by studies conducted with social workers employed in the healthcare field [41,42,43].

In the Italian context, there is still a lack of research on the matter. In order to fill this gap, the present study has investigated the experiences of doctors and nurses working in healthcare systems with specific experience with people affected by cancer, dementia, amyotrophic lateral sclerosis (ALS), and other severe illnesses with respect to their points of view on AHDs, Law No. 219, and euthanasia.

## 2. The Study

### 2.1. Aims and Objectives

The main aim of the study was to detect the relationships among social desirability and extrinsic/religiosity, representations of death, death anxiety, and attitudes toward euthanasia already considered in the international literature [44,45]. Concerning the qualitative part, the main objective was to explore participants’ opinions and concrete experiences with AHDs and to investigate different points of view and experiences in EOL in greater depth.

### 2.2. Participants

Participants were recruited by avalanche sampling in a north-eastern region of Italy. The group was composed of 102 healthcare professionals (63 females and 39 males; 35% physicians, 65% nurses), aged between 22 and 67 years (mean age: 39.6; SD: 13), and working in the healthcare system of the north-east of Italy in departments treating severe illnesses (palliative care 25%, surgery 11%, general medicine 10%, neurology/psychiatry 8%, intensive care and resuscitation 8%, other 38%). Some of them (40%) had been working in their area for less than five years, while 39% had been working in their area for a period ranging from five to 30 years. All the participants, since they were all healthcare professionals, shared the same medium–high economic and social status, and they were all graduated (in medicine or in nursing, according to their profession). Most of them (65%) were religious, among whom 43% actively participated in church activities.

As regards the participants’ recruitment, the researchers first contacted some healthcare professionals they directly knew and who were interested in the study. These first participants were subsequently asked to contact, in turn, if they wanted and could, other physicians or nurses they knew (colleagues or friends) and who could be willing to take part in the research. This way, a vast net of participants was created following a “snowball sampling”, that is, starting from very few participants, a much higher number could be reached progressively thanks to the participants’ intervention.

### 2.3. Method

The study followed a mixed-method design, that is, it undertook both a quantitative point of view, through the use of specific scales, and a qualitative one, in order to explore the participants’ opinions concerning the vast and faceted theme of AHDs in even greater depth [46]. More specifically, a questionnaire was especially developed by the researchers in order to combine these two aspects. The first part of the questionnaire (the quantitative one) was composed of five scales: Attitudes toward Euthanasia (ATE) [47], chosen with the aim of exploring how people consider euthanasia; the Religious Orientation Scale (ROS) [48] to explore the respondents’ religiosity, both intrinsic and extrinsic; the Balanced Inventory of Desirable Responding (BIRD) 16 [49], used to measure the possible influence of social desirability on the participants; the Testoni Death Representation Scale (TDRS) [50,51] to explore the ontological representation of death (annihilation vs. a passage); the Collett–Lester Fear of Death Scale [52] in order to explore participants’ anxieties regarding death.

The second part of the questionnaire (that is, the qualitative section) was composed instead of four open questions inherent to personal consideration of AHDs, changes in work activity caused by Law No. 219, problems and critical incidents related to AHDs, and behavior in cases of requests for euthanasia written in AHDs. As has already been mentioned, the aim of this second section was to explore more in depth, from the direct participants’ narrations of their experiences, their particular points of view concerning AHDs by complementing the first quantitative part, which could not investigate the participants’ unique inner worlds in depth.

The questionnaire was administered online. The participants who were interested in the study and who voluntarily participated signed an informed consent, and, after this passage, they could have access to an internet link that led them to the online page of the questionnaire. The study was approved by the Ethical Committee for Research in Psychology of Padova (Italy) No. 77153A5B74365F7267832129F7FEC799.

The statistical analysis of the quantitative part was conducted through the SPSS software. First, the reliability of each scale was computed using Cronbach’s alpha coefficient. After, a total score for ATE, which is a measure of acceptance of euthanasia, was computed; two total scores for ROS were computed, which respectively measured intrinsic and extrinsic religiosity; a total score for BIRD was computed, which is a measure of social desirability; a total score for TDRS was computed, which is a measure of representation of death as annihilation; and four total scores for Collet–Lester Fear of Death were computed, which are, respectively, a measure of fear of death of the self and of others, and a measure of fear of the process of dying of the self and of others. Then, to evaluate the magnitude of each total score, the mean score was compared with the corresponding central point on a Likert scale using one sample t-test. Correlations were examined using the Pearson correlation coefficient, and, finally, a regression model for acceptance of euthanasia was developed.

The qualitative data obtained from the open-ended questions of the questionnaire, that is, the participants’ answers to those questions, were transcribed verbatim into a text file (one for each participant) and analyzed using thematic analysis, which allowed texts to be examined in terms of their principal contents [53]. The process was performed through the software for qualitative analysis Atlas.it [54], following six main phases: engaging in preparatory organization; reading the texts deeply; coding data; interpreting themes; searching for alternative explanations; and producing the final report [55]. This allowed recurrent concepts that appeared to be particularly meaningful for a participant in every text to be highlighted and to be confronted with those that could be found in other participants’ narrations so that the elements that were shared the most were then grouped together to form some broader thematic categories (the ones that will be presented in the dedicated subsection in the Results) [55].

## 3. Results

### 3.1. Quantitative Data

All study variables show good reliability values and a mean score with a significant departure from the central point of the Likert scale (see Table 1). Healthcare professionals have low scores on acceptance of euthanasia, on both intrinsic and extrinsic religiosity, on fear of death of the self, and on social desirability. To the contrary, they have high scores on representation of death as annihilation, on fear of death of others, and on fear of the process of dying both of the self and of others.

Acceptance of euthanasia shows a significant negative correlation with both intrinsic and extrinsic religiosity (respectively, r = −0.35, *p* = 0.001 and r = −0.22, *p* = 0.031) and a close-to-significant positive correlation with representation of death as annihilation (r = 0.17, *p* = 0.096). Representation of death as annihilation presents a significant negative correlation with both intrinsic and extrinsic religiosity (respectively, r = −0.57, *p* < 0.001 and r = −0.38, *p* < 0.001) and a close-to-significant negative correlation with social desirability (r = −0.19, *p* = 0.064). Fear of death shows only a close-to-significant positive correlation between fear of death of others and extrinsic religiosity (r = 0.19, *p* =.064).

The final regression model for acceptance of euthanasia, including intrinsic and extrinsic religiosity, death representation, fear of death, and social desirability, explains a significant portion of variance (R-squared = 0.24, *p* = 0.022). In particular, we can observe (Table 2) a significant negative impact of intrinsic religiosity (beta = −0.45, *p* = 0.006) and of fear of death of others (beta = −0.38, *p* = 0.011), as well as a significant positive impact of fear of dying of others (beta = 0.32, *p* = 0.024). Consequently, healthcare professionals with a more favorable attitude towards euthanasia are those with a lower intrinsic religiosity and a lower fear of death of others, but with a higher fear of the process of dying of others.

A totally negative opinion concerning AHDs was given by 6% of participants, who considered the law incomplete or inaccurate (67% Catholics, 33% secularists).

### 3.2. Qualitative Data

From the content analysis of the texts, four fundamental thematic areas emerged: “Positive aspects of the new law and of AHDs”; “Negative aspects of the new law and of AHDs”; “Changes occurring in the professional context and critical incidents”; and “Attitudes towards euthanasia requests.”

#### 3.2.1. First Thematic Area: “Positive Aspects of the New Law and of AHDs”

Among all participants, 82% gave a positive opinion about the law, and within this group, 60% were believers and 40% were secularists. With respect to the strengths of Law No. 219 and of the consequent introduction of AHDs into the medical system, almost all participants expressed their positive judgment because AHDs make it possible to safeguard a patient’s self-determination, as Jennifer, a 57-year-old Catholic nurse working in home care services, reported: “The most important strength of the law is the possibility people have to express their will on very delicate matters, such as the end of life.” Agreeing with Jennifer, Pietro, a 38-year-old atheist doctor and the medical director of a surgical ward, said: “This law gives patients the possibility to share their decisions and then to perceive themselves as persons who can decide, and this relieves, even though only a little, the sense of helplessness that being seriously sick causes.” Indeed, AHDs represent for the patients a source of “reassurance for the future, especially for those who present a fatal, terminal pathology”, as Maria, a 24-year-old atheist nurse who worked in a retirement home, reported. This, as Mary further highlighted, also has a positive impact on the patient’s family, since they feel a much less heavy burden because they are not left alone with the necessity to make a decision concerning their loved one’s health and life, but instead, they can have the fundamental support of written directives: “Directives are useful to ‘reduce the responsibility’ of taking certain difficult decisions from the relatives.” All this is possible because the law promotes the patient-centered perspective, as Luisella, a 49-year-old Catholic nurse who was part of a palliative and home care unit, underlined: “The law highlights the importance of the therapeutic alliance with the attending physician, and most of all, that the relationship in itself is already a moment of treatment. This means that physicians and patients have to work together in decision-making processes for the best treatment.” Law No. 219 was considered a very good answer to an urgent necessity, as Giampiero, a 56-year-old man working as a nurse coordinator of an inpatient ward, affirmed: “Before the law, everything was really confusing. And this was very stressful. Now it is possible to cure without anxiety. However, it is necessary that patients write this document.” Ester, a 59-year-old atheist nurse working in a central sterile services department (CSSD) for the sterilization for medical devices and equipment, in this respect, pointed out that “AHDs are useful when a doctor or a nurse has to decide how to act. The law makes their job simpler from the ethical point of view.”

#### 3.2.2. Second Thematic Area: “Negative Aspects of the New Law and of AHDs”

Among the negative aspects that participants underlined was that AHDs are still scarcely known by both the Italian population in general and specifically by healthcare professionals. This was the opinion of 14% of participants, and of Mary, who stated: “There is not enough information for the population; there are no sufficient training events for the healthcare professionals who should deliver clear information to the patients.” Among participants, 7% had received sufficient information concerning AHDs thanks to the healthcare system, while 6% were not interested in this issue or in Law No. 219. Related to this criticality, many participants also raised some doubts concerning the positive effect of the law because a patient could potentially write an AHD without being aware of the implications with respect to the medical procedures involved. In this regard, Paolo, a 28-year-old secular doctor working as a freelancer, said: “AHDs as they are intended by the law can be written in an absolutely vague, raw form, and it would be better if, instead, some pre-compiled forms were available to people. Really, it is not yet clear how to change this aspect of the procedure.” Another negative aspect concerned how professionals could accurately respect patients’ AHDs, since it is difficult to retrieve these documents if patients do not bring the documents with them at the time of hospitalization. This can be very challenging, as Clara, a 48-year-old secular doctor and representative of a surgical ward, reported: “The law does not take into consideration the necessity of healthcare professionals who are assisting people in emergency situations to make some decisions rapidly. How can a doctor imagine the will of patients if they do not have their document with them?” Another potential critical aspect of AHDs was identified in “the influence other people can have on the patients’ capability to express their own choices”, as Carlo, a 31-year-old Catholic nurse working in a surgical unit of a hospital, said: “Sometimes, severely ill patients are really unable to understand and then to choose the treatment. Socially speaking, it’d be necessary to prepare people before they are ill.” Particular attention was paid to the difficulties related to the Catholic Italian culture, which is generally perceived as contrary to Law No. 219 and, therefore, socially and politically committed to opposing effective law enforcement, as Mary explicated: “Since this new law has just started to be implemented, in a country in which medical, popular, and scientific culture are strongly affected by the Church, we will need an intense and long-term effort. Unfortunately, the Church really interferes in the health system, and this causes lots of problems.” Anna, a 35-year-old secular nurse working in a neurosurgery ward, agreed with this, saying: “A critical point is our own mentality, which is still obsolete with regard to certain matters. Traditional and authoritarian perspectives are still alive in our health system”, while Lisa, a 37-year-old Catholic nurse working as part of a palliative home care unit, specified that “In Italy, we really need a cultural change. It is necessary if we want to apply this law and work better.”

#### 3.2.3. Third Thematic Area: “Changes Occurring in the Professional Context and Critical Incidents”

The changes brought by Law No. 219 were indicated especially by professionals who have worked in departments with EOL patients and in emergency rooms. The first area inhered to the communication between the healthcare professionals and patients, which was generally reported to be more open and qualitatively improved, as Giancarlo, a 37-year-old secular doctor working as the director of a surgical ward, described: “I have noticed more open discussion between patients and their relatives, especially concerning the details of the law and ways to obtain the related documentation. Somehow, we have to learn how to manage this change because we are not yet prepared enough for this task.” The importance of communication especially involves relationships with family members: “Often, relatives have to make a decision when their beloved one suffers from dementia. It is very important to be able to explain everything clearly, because in the majority of cases, they have no written document to prove the real will of the patient. It is important to allow them to make the choice they consider the best. In this way, in the majority of cases, they go so far as to refuse disproportionate treatment.” However, communication difficulties have also appeared in healthcare teams, as narrated by Lisa: “There was a patient with heart failure at the final stage, complicated by a stroke; a relative, during the patient’s admission, asked to let the person reach his end of life naturally, without further interventions, which was something the patient himself had previously expressed. The attending physician agreed, but the physician who came after him didn’t, so he positioned a nasogastric tube for the patient’s nutrition and transferred him from intensive care into the neurologic ward, where he was monitored and also tied up because he was agitated. Of course, the patient tore away his tube. He was not sedated at all until his relatives decided to transfer him to a hospice, where he died peacefully two days later.”

#### 3.2.4. Fourth Thematic Area: “Attitudes Towards Euthanasia Requests”

If expressed by the AHDs, euthanasia would be practiced by 24% of participants (48% Catholics, 48% secular, 4% agnostics), while 37% are against euthanasia, among whom 55% said “because it is illegal” (67% Catholics; 33% secular); 20% stated that they do not have an opinion and that they would abide by the decisions of the team or the patient’s family members (62% Catholics; 33% secular, 5% agnostics); 19% did not answer the question (65% Catholics; 35% secular).

Among both groups, there were those who would not practice euthanasia. The refusal of euthanasia was generally accompanied by the participants’ will to be able to support the patient as best as they could, as Stella, a 23-year-old Catholic nurse working in a retirement home, explained: “I think that there are still many deficiencies in the knowledge people have of palliative care, and I believe that this gap could lead someone to think that it is possible to choose euthanasia. I always listen to patients who want to die, their thoughts and worries, but I also clarify that in Italy, it is not possible. Also, sometimes people are convinced that terminal sedation is equivalent to euthanasia. This is incorrect. Italian people really need to know better what palliative care and anticipated directives are.” On the other hand, the correct practice of palliative care was considered as an efficacious remedy against the requests for euthanasia, as Mina, a 29-year-old secular nurse working in a retirement home, emphasized: “Everything depends on the context. Really, I can help patients to manage end of life with palliative care, and in this way I would certainly do whatever it takes to make sure any request of the patient is fulfilled, although sometimes, this may not totally match the patients’ will, since they want to die despite the palliative care intervention.” Among other Catholics, Luisella spoke of “explaining to a person that euthanasia in Italy is not legal, and that they should go abroad”; Adamo, a 33-year-old general practitioner, said “I believe I would not comply with the patient’s requests, since euthanasia is not necessarily linked to pathologies for which there are no effective therapeutic solutions”; Antonio, a 55-year-old anesthesiologist, said “I could not practice it.” Among secularists, Amanda, a 24-year-old nurse in general surgery, said “My professional actions are also bound by laws that wouldn’t allow me to practice euthanasia, regardless of my ethical and moral beliefs.”

To the contrary, other participants, both Catholics and secularists, would practice euthanasia because they first consider the fulfilment of their patients’ requests as a fundamental task of their clinical practice, and then, secondarily, their serious ethical and practical difficulties in respecting them totally. Among the secularists who believe that euthanasia can be practiced are Gabriele, a 30-year-old nurse, who said, “I would encourage the doctor to respect them and to make sure they are respected”; Giuliana said she would approach it “with serenity, accepting without prejudice the patient’s decision.” Among the Catholics who would practice euthanasia are Roberto, specializing in neurosurgery for 30 years: “I would accept the will of the patient”; Lucia, a 24-year-old nurse: “In my opinion, the individual’s will is superior to every personal conviction. I would accept the request as if it was mine: If a patient asks for euthanasia, it is correct to respect his/her request”; and Jessica, a 53-year-old nurse working in neurosurgery: “Personally, I would respect it; I am not against euthanasia […]. The will of the individual is, above all, a personal opinion. I would accept the request as if it were my own”; Giusy, a 44-year-old nurse: “I would put into practice the intentions expressed by the patient.”

Marco, a 23-year-old Jewish nurse working in the infirmary of a prison, said: “I would seek the advice of someone who is more expert than me in the matter […] I would request the assistance of the Ethical Committee.” Similarly, 6% of participants declared themselves as unable to manage these issues because they were not competent on Law No. 219, like Luca, a 60-year-old Catholic doctor and director of a surgery ward: “I have never studied the topic in detail. For the moment, I do not feel the necessity to do it, neither from a personal point of view nor from a professional one. I don’t care about it.”

## 4. Discussion

The quantitative results, highly consistent with the starting hypotheses, are in line with the literature [11] and with the principles of the Terror Management Theory [13,18]. Indeed, the negative correlation between religiosity and the participants’ attitudes towards euthanasia was confirmed, together with a positive correlation between social desirability and religiosity, that is, religious participants represent death as a passage and do not accept euthanasia. To the contrary, those who have more positive attitudes towards euthanasia represent death as annihilation. More specifically, as the final regression model for acceptance of euthanasia shows, lower intrinsic religiosity reduces fear of death of others, but enhances fear of the process of dying of others. It is important to remember that there is a difference between the fear of death and the fear of witnessing dying. In fact, witnessing dying implies dealing with the suffering of those who die, and this is the case of healthcare professionals with a lower intrinsic religiosity. On the other hand, the analysis also shows that participants with a lower intrinsic religiosity and a lower fear of death of others but with a higher fear of dying of others have more favorable attitudes towards euthanasia. This initial result seems to show that, compared to the fear of the suffering of dying of others, euthanasia is not a solution for those who believe, while, on the contrary, it is for those who do not believe.

However, this result, which seems so in line with the literature and the most common forms by which Italians are represented with respect to the problem of euthanasia, does not perfectly collide with the further results obtained for the qualitative part. The analyses of the texts showed that, for the most part, Catholics had positive attitudes toward Law No. 219. This result confirms that there is a wide segment of the Catholic world that does not agree with the traditional position of the Church, which severely contrasts approval and the application of the law. Furthermore, a quarter of all the participants, equally divided between believers and non-believers, would be willing to practice euthanasia if this choice were expressed in an AHD.

Emerging among the reasons for positively considering Law No. 219 was the possibility for patients to achieve self-determination concerning their health and medical treatments, as Gianna and Pietro testified. This is in line with the literature, which reports how the right to self-determination is a key concept considered related to the idea of dignity and freedom of choice in the end of life [56,57]. This law was indicated as an initial solution to fill a gap in the Italian scenario concerning medical informed consent, which caused no-longer-bearable uncertainty in everyday health practices. Texts pay particular attention to the Italian cultural and social context, in which the influence of the Catholic Church and of the religious mentality was considered as a serious obstacle to the application of Law No. 219, as Maria and Lisa affirmed, and has already been emphasized in the literature [16]. As Giancarlo said, in line with the literature [58], the management of AHDs modifies the relationships among patients, families, and the health community, particularly regarding physicians, and this may cause problems. Indeed, ideologies and religions often influence moral perspectives and behaviors, especially when one has to face death [59]. This is why the traditional authoritative doctor–patient model has not totally passed and new behavioral relational schemes have not yet been assumed.

With respect to the possible confusion between Law No. 219 and euthanasia, it emerged that even though the attitudes towards euthanasia were mostly negative, many participants were persuaded that it is possible to respect this request by some patients. A possible explanation of this could be once again related to imprecise knowledge of the meaning of euthanasia and what AHDs permit on the basis of Law No. 219. It may be that this confusion will still create some negative reactions with respect the total acceptance of the same law. However, a final comment must be made with respect to the criticism made by some participants regarding the applicability of the law given the difficulties of accessing a patient’s document, because the patient may have handed it over to the municipality and not have it with him/her at the time of admission. In fact, AHDs can be drawn up in various forms, such as a public deed or a notarized private deed delivered personally by the patient at the office of the municipality of residence, which then records it in a special register, where it is established. AHDs can also be delivered personally to the health facilities with the indication of a trustee. Currently, the problem of accessibility is already being solved, as in the budget law of 2018, two million euros were allocated for the preparation of a national database that makes these documents accessible to all health institutions associated with the national health system. With the subsequent decree of 10 December 2019, No. 168, the method of collecting copies of AHDs in the same national database at the Ministry of Health was further regulated, with additional funding of 400,000 euros.

The lack of psychological training in order to reduce death anxiety may be one of the causes of this phenomenon. Furthermore, all the potential implications of the law are not yet appropriately and completely known, so professionals need to participate in training events, as Maria reported. There were even some participants who actually knew nothing about these issues, as expressed by Luca. This problem is already stressed in the literature [36,37,38], and it is particularly significant because of its legal implications and the authentic respect for the patient-centered approach, which characterizes respect for the self-determination principle. In this perspective, palliative psychologists could offer important support to health professionals, offering them the opportunity to integrate legal and relational competencies with respect to EOL decision-making competencies [2].

All these difficulties related to the lack of relational and communication competences could be managed with appropriate death education courses, which would allow people to face anxiety and uncertainties deriving from both legal and death-related issues [60]. Death education courses that palliative psychology can make available to patients, family members, and health practitioners can, in fact, through open and frank dialogue, address and resolve the ethical uncertainties and differences in ideological positions from which misunderstandings and conflicts arise in order to ensure the most appropriate way to respect the will of the patient [2].

## 5. Conclusions

The present research confirms the fundamental importance in Italy of the Law No. 219/2017 and the related introduction of AHDs into the healthcare system to finally enable patients and their relatives to talk about AHDs with health professionals they trust. However, in order to become reliable references, healthcare professionals are supposed to be able to face this task with appropriate legal competence and a high level of psychological skill, allowing the end of life to be what the patient wants, instead of what is more profitable for the healthcare system or for health professionals.

In this regard, however, the study has also highlighted some criticalities reported by health practitioners, among which the most important one is the lack of adequate knowledge concerning AHDs, which can, in turn, lead to some confusion about the concept of euthanasia, creating a serious backlash to the development of individualized and attentive care combined with ethically and clinically correct choices. Despite all this, both physicians and nurses who took part in the research expressed positive attitudes towards AHDs, without even a significant difference between religious believers and secularists.

Indeed, palliative psychologists may offer important interventions to help manage the decision-making process while also focusing on a proper death education program in order to support the elaboration of the bad news and to enable patients and healthcare professionals to create a shared care plan that also includes AHDs.

### Limitations of the Study and Future Perspectives

This research has some limitations, and a fundamental one, as has already been highlighted elsewhere, lies in that it seems that the application of a specific scale, the Collett–Lester Fear of Death Scale, may not be completely appropriate for studies in this field, since all the measures related to the scale provided some counterintuitive results. This had an impact on the possibility of combining certain qualitative results with the quantitative ones, since they differed. For the future, it could be therefore useful to choose instead the Death Anxiety Scale, which is more frequently adopted by researchers.

Furthermore, since the results obtained with the Collett–Lester Fear of Death Scale [52] did not permit detection of correlations among religiosity, social desirability, representation of death, and death anxiety, in future studies, it could be useful to replace this scale with the Death Anxiety Scale [61].

Moreover, it would be important to check how and whether time may influence changes in the Italian cultural perspective on this issue by exploring the different physicians’ and nurses’ attitudes towards AHDs and their competences in this regard. This could offer even more insight into the criticalities already described in the present research and guide the implementation of proper training on AHDs, together with some fundamental death education courses for both healthcare professionals and the whole community as well.

## Figures and Tables

**Table 1 behavsci-10-00160-t001:** Reliability and descriptive statistics for study variables with comparison of mean scores with the central point of a Likert scale using a *t*-test.

Variable	Cronbach’s Alpha	Range	M	SD	*t*	*p*-Value
ATE—Acceptance of euthanasia	0.86	1–5	2.66	0.91	−3.81	<0.001
ROS—Intrinsic Religiosity	0.94	1–5	2.48	1.14	−4.44	<0.001
ROS—Extrinsic religiosity	0.80	1–5	2.38	0.72	−8.30	<0.001
TDRS—Death as Annihilation	0.87	1–5	3.48	1.03	4.76	<0.001
CL—Fear of Death of Self	0.77	1–5	2.98	0.92	−0.25	<0.001
CL—Fear of the Process of Dying of Self	0.82	1–5	3.67	0.84	8.10	<0.001
CL—Fear of Death of Others	0.75	1–5	3.95	0.71	13.70	<0.001
CL—Fear of the Process of Dying of Others	0.83	1–5	3.33	0.91	3.71	<0.001
BIRD—Social Desirability	0.67	1–7	4.33	0.72	−9.49	<0.001

**Table 2 behavsci-10-00160-t002:** Regression analysis of religiosity, death representation, fear of death, and social desirability on acceptance of euthanasia.

Variable	Beta	*t*	*p*-Value
ROS—Intrinsic Religiosity	−0.45	−2.85	0.006
ROS—Extrinsic religiosity	0.10	0.71	0.478
TDRS—Death as Annihilation	−0.07	−0.61	0.544
CL—Fear of Death of Self	0.18	1.43	0.157
CL—Fear of the Process of Dying of Self	−0.14	−1.05	0.296
CL—Fear of Death of Others	−0.38	−2.59	0.011
CL—Fear of the Process of Dying of Others	0.32	2.29	0.024
BIRD—Social Desirability	0.15	1.47	0.147

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
