# Peer review of "A Challenge for Palliative Psychology: Freedom of Choice at the End of Life among the Attitudes of Physicians and Nurses"

_behavsci, 2020, doi:10.3390/bs10100160_

Round 1
Reviewer 1 Report
The present study by Testoni et al. titled " A challenge for palliative psychology: Freedom of choice at the end of life among the attitudes of physicians and nurses." A well written and well-investigated article.
However, my suggestions to authors
1) Add a table about the participants, describing their social, economic, and educational status.
2) Conclusion is too long.
3) Add a future perspective.
Author Response
Dear Reviewer,
We thank you sincerely for the possibility to revise our article “A challenge for palliative psychology: Freedom of choice at the end-of-life among the attitudes of physicians and nurses”, manuscript ID: behavsci-957758.
The article that we send you again, for the possible publication on your journal “Behavioral Sciences”, has now been completely revised on the basis of your indications.
In the following section of this letter, all your indications will be reported, and for each of them a description of the specific operations we conducted in our text in order to follow the given suggestions will be provided. All the changes in the manuscript have been written in the color red, in order to make them easier to detect.
In order to ensure that this section of the letter could be as clear as possible, we have created a simple table that reports, in the first column, all your indications, one by one, and in the second column our response to each of them.
|
Reviewer #1 Comments/Requests
|
Authors’ response |
|
The present study by Testoni et al. titled " A challenge for palliative psychology: Freedom of choice at the end of life among the attitudes of physicians and nurses." A well written and well-investigated article.
However, my suggestions to authors
1) Add a table about the participants, describing their social, economic, and educational status.
|
For the present research the participants’ specific economic and social status was not directly assessed, since all the participants belonged to the same social category (healthcare professionals, and more specifically physicians and nurses), and they all worked and lived in similar conditions and in the same geographical area (North-East of Italy). The participants’ educational status was homogeneous as well, since they were all graduated professionals.
Because of this, the researchers do not believe it could be useful to create a table with this information, which was not directly explored and is homogeneous for all the participants. A sentence concerning the participants’ social, economic and educational status has however been inserted in the dedicated section of the manuscript (on page n. 3 lines 136-138). |
|
2) Conclusion is too long. |
As indicated, the Conclusion section of the manuscript has been revised and summarized, in order to make it clearer and shorter. The new version of this section can be now found on pages n. 10-11 of the manuscript. |
|
3) Add a future perspective. |
As required, suggestions concerning future perspectives have been inserted in the text, after the Conclusion section, in the section that has been renamed “Limitations of the study and future perspectives”, on page n. 11, lines 466-471. |
We sincerely hope that the described modifications and additions might represent a valuable and exhaustive answer.
Thank you again for this opportunity and for your consideration of this new version of the manuscript.
Reviewer 2 Report
This research is interesting, and I enjoyed reading it. Thank you so much.
Please consider my comments as a possible aid towards more conciseness in the presentation of your work.
Lines 125-130:
Regarding the first paragraph of the Methods section, please confer references number 40 to 46; possibly, they are one number ahead of the number described in the references section.
Lines 566-567:
Instead: (…) Physician Attitudes Toward Advanced Directives. (…)
Proposal: (…) Physician Attitudes Toward Advanced Directives: A Literature Review of Variables Impacting on Physicians Attitude Toward Advance Directives. (…)
Lines 221:
You stated: Negative opinions were given by 6% of participants, (…)
Lines 222-224:
You stated: (…) Among the negative aspects that participants underlined was that AHDs are still scarcely known by both the Italian population in general and specifically by healthcare professionals. This was the opinion of 14% of participants, (…)
Could you please specify these numbers better? It seems, for the reader, that more than 6% of the participants had negative opinions towards AHDs.
Lines 278-282:
Please check percentages (24%, 37%, 20% and 20%).
Line 281:
You stated: (…) 62% Catholics; 33% secular (…)
Were there 5% of agnostics in this group?
Lines 178-180:
You stated: (…) healthcare professionals with a more favourable attitude towards euthanasia are those with a lower intrinsic religiosity, with a lower fear of death of others but with a higher fear of the process of dying of others.
Lines 331-333:
You stated: (…) More specifically, as the final regression model for acceptance of euthanasia shows, intrinsic religiosity reduces fear of death of others but enhances fear of the process of dying of others.
Proposal: (…) More specifically, as the final regression model for acceptance of euthanasia shows, lower intrinsic religiosity reduces fear of death of others but enhances fear of the process of dying of others.
Please rethink the discussion and conclusions in this regard.
Author Response
Dear Reviewer,
We thank you sincerely for the possibility to revise our article “A challenge for palliative psychology: Freedom of choice at the end-of-life among the attitudes of physicians and nurses”, manuscript ID: behavsci-957758.
The article that we send you again, for the possible publication on your journal “Behavioral Sciences”, has now been completely revised on the basis of your indications.
In the following section of this letter, all your indications will be reported, and for each of them a description of the specific operations we conducted in our text in order to follow the given suggestions will be provided. All the changes in the manuscript have been written in the color red, in order to make them easier to detect.
In order to ensure that this section of the letter could be as clear as possible, we have created a simple table that reports, in the first column, all your indications, one by one, and in the second column our response to each of them.
|
Reviewer #2 Comments/Requests
|
Authors’ response |
|
1) This research is interesting, and I enjoyed reading it. Thank you so much.
Please consider my comments as a possible aid towards more conciseness in the presentation of your work.
Lines 125-130:
Regarding the first paragraph of the Methods section, please confer references number 40 to 46; possibly, they are one number ahead of the number described in the references section.
|
As suggested, all the references in the text have been checked, in order to verify whether their number was consistent with the references listed at the end of the manuscript.
The references in the text that resulted to be wrongly numbered have therefore been corrected, and can be seen written in color red.
|
|
2) Lines 566-567:
Instead: (…) Physician Attitudes Toward Advanced Directives. (…)
Proposal: (…) Physician Attitudes Toward Advanced Directives: A Literature Review of Variables Impacting on Physicians Attitude Toward Advance Directives. (…)
|
As indicated by the reviewer, the title of the cited article has been modified, and the missing part of it has been added.
The current version of the reference in the text is therefore:
“37. Coleman, A. M. Physician Attitudes Toward Advanced Directives: A Literature Review of Variables Impacting on Physicians Attitude Toward Advance Directives. American Journal of Hospice and Palliative Medicine 2012, 30(7), 696-706; doi:10.1177/1049909112464544.” |
|
3) Lines 221:
You stated: Negative opinions were given by 6% of participants, (…)
Lines 222-224:
You stated: (…) Among the negative aspects that participants underlined was that AHDs are still scarcely known by both the Italian population in general and specifically by healthcare professionals. This was the opinion of 14% of participants, (…)
Could you please specify these numbers better? It seems, for the reader, that more than 6% of the participants had negative opinions towards AHDs. |
As regards the given percentages, the first one (6%) is related to the quantitative data, more specifically it concerns the percentage of participants who had only a negative opinion about AHDs and could not find anything positive about them.
The second percentage, on the contrary, concerns instead the qualitative data, and is related to the specific question “Which are, in your opinion, some positive and some negative aspects of AHDs?”. Since all the participants had to answer to this question, even some of them who considered AHDs to be indeed very important and had a positive attitude towards them gave their opinion concerning the possible presence of negative aspects, because a reflection concerning these themes was required by the question itself, and this is why the percentage here is higher (14%).
In order to make this difference clear and to avoid any confusion, the indication concerning the first percentage (6%) has been moved to the Quantitative Data subsection of the Results, where it belongs (page n. 6 lines 219-220). |
|
4) Lines 278-282:
Please check percentages (24%, 37%, 20% and 20%).
|
As suggested, these given percentages have been checked, and since a mistake had occurred concerning one of them (the last one of the four presented, related to those participants who did not answer the question), the correct number has been inserted. The correct percentages are therefore: 24%, 37%, 20% and 19%. The change can be found on page n. 8 line 321. |
|
5) Line 281:
You stated: (…) 62% Catholics; 33% secular (…)
Were there 5% of agnostics in this group? |
As suggested, the sentence has been checked and the correct percentage of agnostics had been omitted by mistake. There was therefore a 5% of agnostic people in that group (the information has been added in the text, on page n. 8 line 320).
|
|
6) Lines 178-180:
You stated: (…) healthcare professionals with a more favorable attitude towards euthanasia are those with a lower intrinsic religiosity, with a lower fear of death of others but with a higher fear of the process of dying of others.
Lines 331-333:
You stated: (…) More specifically, as the final regression model for acceptance of euthanasia shows, intrinsic religiosity reduces fear of death of others but enhances fear of the process of dying of others.
Proposal: (…) More specifically, as the final regression model for acceptance of euthanasia shows, lower intrinsic religiosity reduces fear of death of others but enhances fear of the process of dying of others.
Please rethink the discussion and conclusions in this regard. |
The suggested modification has been inserted, since the new version of the statement, indicated by the reviewer, was indeed more correct and appropriate, and all the text has been revised and modified accordingly where needed (page n. 9). |
We sincerely hope that the described modifications and additions might represent a valuable and exhaustive answer.
Thank you again for this opportunity and for your consideration of this new version of the manuscript.
Reviewer 3 Report
This is an interesting study that follows on from others in the field, cross-disciplines and contexts. The authors provide a sound study with methodological rigour. There are two areas that need improvement before this has the applicability that the authors aspire.
First, further details are needed in the methodology, especially regarding participant recruitment. The method and process of the study are not absolutely clear at the moment.
Next, the paper is clearly taking a psychological view on the matter - whether this is due to the authors; backgrounds or the literature used. There need to be a wider discussion and information in the background section, on this subject, drawing on an extensive body of literature across scientific areas like thanatology, social work and nursing that have largely contributed to the discourse altogether. Currently, the paper is merely contextualised in a linear way, that narrows its readership as well.
Author Response
Dear Reviewer,
We thank you sincerely for the possibility to revise our article “A challenge for palliative psychology: Freedom of choice at the end-of-life among the attitudes of physicians and nurses”, manuscript ID: behavsci-957758.
The article that we send you again, for the possible publication on your journal “Behavioral Sciences”, has now been completely revised on the basis of your indications.
In the following section of this letter, all your indications will be reported, and for each of them a description of the specific operations we conducted in our text in order to follow the given suggestions will be provided. All the changes in the manuscript have been written in the color red, in order to make them easier to detect.
In order to ensure that this section of the letter could be as clear as possible, we have created a simple table that reports, in the first column, all your indications, one by one, and in the second column our response to each of them.
|
Reviewer #3 Comments/Requests
|
Authors’ Response |
|
1) This is an interesting study that follows on from others in the field, cross-disciplines and contexts. The authors provide a sound study with methodological rigor. There are two areas that need improvement before this has the applicability that the authors aspire.
First, further details are needed in the methodology, especially regarding participant recruitment. The method and process of the study are not absolutely clear at the moment. |
As suggested, the Method section of the manuscript has been revised, in order to explain in more detail all the passages undertaken during the implementation of the research. More specifically, the participants’ recruitment has been accurately described (on pages n. 3-4, lines 140-146), together with the precise method and process adopted for the study (pages n. 4 and 5). |
|
2) Next, the paper is clearly taking a psychological view on the matter - whether this is due to the authors; backgrounds or the literature used. There need to be a wider discussion and information in the background section, on this subject, drawing on an extensive body of literature across scientific areas like thanatology, social work and nursing that have largely contributed to the discourse altogether. Currently, the paper is merely contextualized in a linear way, that narrows its readership as well. |
As suggested, the Introduction section of the manuscript has been revised in order to broaden the perspective on AHDs, adding also the insights offered by studies in the field of Thanatology, as well as ones from the point of view of nurses and social workers (on pages n. 2 and 3). |
We sincerely hope that the described modifications and additions might represent a valuable and exhaustive answer.
Thank you again for this opportunity and for your consideration of this new version of the manuscript.
Round 2
Reviewer 2 Report
I have no more comments to make.
Congratulations and all the best for your paper.
Reviewer 3 Report
The comments have been addressed sufficiently.